# From dawn till dusk: Time-adaptive bayesian optimization for neurostimulation

**John E. Fleming**[1]◉*, **Ines Pont Sanchis**[2]◉, **Oscar Lemmens**[2], **Angus Denison-Smith**[2], **Timothy O. West**[1,3], **Timothy Denison**[1,2]◉, **Hayriye Cagnan**[1,3]◉*

**1** Medical Research Council Brain Network Dynamics Unit, Nuffield Department of Clinical Neurosciences, University of Oxford, Mansfield Road, Oxford, United Kingdom, **2** Institute of Biomedical Engineering, Department of Engineering Science, University of Oxford, Old Road Campus Research Building, Oxford, United Kingdom, **3** Department of Bioengineering, Imperial College London, White City Campus, London, United Kingdom

◉ These authors contributed equally to this work.
* h.cagnan@imperial.ac.uk; john.fleming@ndcn.ox.ac.uk

## Abstract

Stimulation optimization has garnered considerable interest in recent years in order to efficiently parametrize neuromodulation-based therapies. To date, efforts focused on automatically identifying settings from parameter spaces that do not change over time. A limitation of these approaches, however, is that they lack consideration for time dependent factors that may influence therapy outcomes. Disease progression and biological rhythmicity are two sources of variation that may influence optimal stimulation settings over time. To account for this, we present a novel time-varying Bayesian optimization (TV-BayesOpt) for tracking the optimum parameter set for neuromodulation therapy. We evaluate the performance of TV-BayesOpt for tracking gradual and periodic slow variations over time. The algorithm was investigated within the context of a computational model of phase-locked deep brain stimulation for treating oscillopathies representative of common movement disorders such as Parkinson's disease and Essential Tremor. When the optimal stimulation settings changed due to gradual and periodic sources, TV-BayesOpt outperformed standard time-invariant techniques and was able to identify the appropriate stimulation setting. Through incorporation of both a gradual "forgetting" and periodic covariance functions, the algorithm maintained robust performance when a priori knowledge differed from observed variations. This algorithm presents a broad framework that can be leveraged for the treatment of a range of neurological and psychiatric conditions and can be used to track variations in optimal stimulation settings such as amplitude, pulse-width, frequency and phase for invasive and non-invasive neuromodulation strategies.

## Author summary

Brain stimulation is an effective intervention for medically refractory neurological and psychiatric disorders. Widespread clinical usage is however held back by the time burden required to determine optimal patient-specific parameters which currently relies on an

**Data Availability Statement:** The source code used to produce the results and analyses presented in this manuscript is available for download from the OUI Software Store (https://

process.innovation.ox.ac.uk/software/) with reference 21242.

**Funding:** This work was supported by the Medical Research Council UK Award (MR/R020418/1 to HC); the Wellcome Institutional Strategic Support Fund (ISSF) (grant ref 204826/Z/16/Z to HC, TD and TW) and the UK Medical Research Council (MC_UU_00003/3 to TD and JF). The funders had no role in study design, data collection and analysis, decision to publish, or preparation of the manuscript.

**Competing interests:** We have read the journal's policy and the author(s) of this manuscript have the following competing interests: The University of Oxford has research agreements with Bioinduction Ltd to explore circadian stimulation methods. TD has business relationships with Bioinduction for research tool design and deployment, and stock ownership (< 1%). TD is a Founder and Director of Amber Therapeutics Ltd which also has a controlling stake in Bioinduction Ltd and Finetech Medical Ltd. JEF, OL, AD-S, HC, TD are inventors on a pending patent application related to the subject matter of this paper. IPS and TOW have no conflicts of interest to declare.

empirical process of trial and error. Additionally, clinical approaches to date have assumed that the mapping between effective stimulation settings and patient's symptom severity is fixed over time. There is increasing evidence however that symptom severity and profile can fluctuate over multiple timescales due to a variety of factors such as medication intake, biological rhythms, and disease progression. Here, we introduce a time varying Bayesian Optimization algorithm to maintain optimal parameters for controlling neurostimulation amidst shifting physiological demands. We provide an in-silico evaluation of this technique using a computational model of synchronous neural activity. Our results demonstrate that the proposed algorithm outperforms static controllers and can simultaneously track gradual and periodic variations in optimal stimulation parameters over time. This provides preliminary evidence that our proposed framework enables dynamic neuromodulation. This approach can be leveraged to improve treatment delivery for complex disorders such as epilepsy and Parkinson's disease for which time varying factors can compromise treatment efficacy.

## 1. Introduction

Over the last four decades neuromodulation has been adopted as an adjunct therapy for a variety of treatment resistant neurological and psychiatric disorders [1–8], whereby electrical or magnetic stimulation is applied to certain brain regions to reduce disease symptoms. Clinical application of invasive and non-invasive stimulation-based therapies (i.e. Deep Brain Stimulation, Transcranial Magnetic Stimulation, etc.) have so far relied on empirical selection of patient-specific stimulation parameters [9,10]. Stimulation is parametrized according to both symptom severity and side-effects, where an ideal parameter set is one that minimizes both. Due to the high dimensionality of the stimulation parameter space, manual identification of effective therapy parameters can be a difficult and time-consuming process [11]. Another drawback, recently acknowledged by the neuromodulation community, are potential limitations associated with titrating therapies based on a brief disease snapshot available from day-time clinical assessments [12].

Optimization and machine-learning techniques have received considerable interest for efficient identification of patient-specific stimulation parameters [13,14]. These techniques often focus on sequential evaluation of samples from the stimulation parameter space to characterize and guide future exploration of the parameter space. Implementation of these approaches in practice requires objective quantification of each sample selected from the parameter space in terms of its effect on patient's symptom severity. Biomarkers (i.e., signals that correlate with specific states of health or disease) can be derived from invasive and non-invasive recordings, and utilized to determine the relationship between therapy settings and performance. A variety of such signals have been explored; for instance synchronous neural activity in the beta band for controlling stimulation in Parkinson's disease and Essential Tremor, and gamma power in treatment resistant depression [15–19].

Although several optimization approaches have been explored in both clinical and computational settings [20–25], a limitation of these is that the therapy parameter space is assumed to be time-invariant (i.e., fixed). In practice, a patient's symptom severity can vary across multiple timescales due to factors, such as changes over several hours due to medication intake [15] or the sleep/wake circadian rhythm [12,26,27]; or fluctuations at a much longer timescale due to disease progression [28]. For this reason, optimization approaches which focus on identification of a time-invariant, or stationary, optimal stimulation parameter set may provide

suboptimal therapy as the patient's symptoms, or the mapping between stimulation settings and performance, drifts in response to these temporal variations.

To address these limitations, we designed a time varying controller that can maintain optimal stimulation performance in the face of shifting physiological demands that might be expected from factors highlighted above. This controller is based upon Bayesian Optimization (*BayesOpt*) with modifications to enable time-adaptive therapy optimization (Time-Varying Bayesian Optimization–*TV-BayesOpt*).

We evaluated the time-varying controller's performance using a population model of synchronous neural activity–coupled Kuramoto oscillators [29]. Kuramoto oscillators and their response to external perturbation, such as deep brain stimulation (DBS), have been used as a model for common movement disorders such as Parkinson's disease, and Essential Tremor [30]. These pathologies, sometimes referred to as oscillopathies [31], exhibit increased neural synchrony and rhythmic activity across disease circuits when patients are symptomatic [32,33] versus suppression of associated neural rhythms when patient's symptoms are effectively managed [15,34,35]. This close relationship licenses our use of a model of neural synchrony and its control via external perturbation as a proxy to understand modulation of clinical symptoms with brain stimulation [36–39]. Performance of the TV-BayesOpt algorithm for tracking periodic drifts, representative of fluctuations due to the sleep/wake or medication schedule, and linear drifts, representative of more gradual variations due to disease progression, were investigated. In contrast to BayesOpt, TV-BayesOpt was continuously able to track variations in the optimal stimulation parameter set and resulted in maintained suppression of the modelled biomarker, minimizing symptom rebound over the optimization process. We demonstrate the utility of our time-varying controller in the context of pathological tremor while optimizing stimulation timing during phase specific DBS. However, our observations and approach can readily be extended to other neurological and psychiatric conditions, underpinned by rhythmic neural activity, and used for the optimization of various stimulation parameters (i.e., amplitude, pulse width, frequency, and phase).

## 2. Materials and methods

### 2.1. Overview of methods

In this section, we detail the model used to evaluate the performance of our time-varying controller whereby external perturbations are optimally parametrized in order to suppress a rhythmic biomarker. We then familiarize the reader with the specifics of the *BayesOpt* algorithm and outline our choices, namely the *surrogate model* (i.e., the model used to estimate the shape of the true mapping between symptom severity and stimulation parameters) and the *acquisition function* (i.e., how to select the next sample from the parameter space). We finally introduce the *spatiotemporal covariance functions* used for a time-varying implementation of the BayesOpt algorithm (*TV-BayesOpt*) that incorporate prior knowledge regarding the form of the objective function along with its anticipated drift in time.

### 2.2. Kuramoto model of a synchronous biomarker and deep brain stimulation

The Kuramoto model provides a simple mathematical model to describe synchronization in biological systems [40]. In a variety of neurological disorders, pathologically increased neuronal synchrony, reflected as an increase in power within a specific frequency band, is observed to correlate with disease symptom severity [15–18]. In this specific case, the Kuramoto model represents a population of neurons as N coupled oscillators that oscillate about some center

frequency, $\omega$. To mimic synchronous activity associated with a specific disease pathology, the value of $\omega$ is selected to lie within a frequency band that is observed to correlate with the severity of the associated disease's symptoms. In this study, to model synchronous oscillatory activity characteristic of Essential Tremor, where tremor is observed to occur between 4–12 Hz, $\omega$ was selected as 8 Hz [41]. The frequency of each oscillator simulated in the population was selected from a Gaussian distribution centered around 8 Hz with a standard deviation of 0.0075. This resulted in a distribution of oscillator frequencies in the population between $8 \pm 0.15$ Hz. In a practical application, $\omega$ would need to be identified in a patient-specific manner. This can be done by using accelerometer recordings from tremor patients to determine the patient specific average tremor frequency and associated frequency variability. Alternatively, for other neurological disorders $\omega$ may be identified using non-invasive and invasive neural biomarkers which can be derived using electroencephalogram (EEG) and local field potential (LFP) recordings. The strength of interaction between the oscillators is captured by a coupling term $\gamma$, where the influence of the $j^{th}$ oscillator on the $i^{th}$ oscillator is described in terms of phase progression (i.e., speeding up or slowing down) of the $i^{th}$ oscillator depending on their difference in phase.

$$\frac{d\theta_i}{dt} = \omega_i + \frac{\gamma}{N} \sum_{j=1}^{N} \sin(\theta_j - \theta_i) \tag{1}$$

To simulate the influence of electrical stimulation such as DBS on the behavior of the population, the model has been extended to account for the effects of exogenous perturbations [42]. This extended model is represented as:

$$\frac{d\theta_i}{dt} = \omega_i + \frac{\gamma}{N} \sum_{j=1}^{N} \sin(\theta_j - \theta_i) + IX(t)Z(\theta_i) \tag{2}$$

where $I$ is the intensity of stimulation, $X(t)$ is a function whose value equals 1 at time $t$ if stimulation is applied and is 0 otherwise, and $Z(\theta_i)$ is the phase response curve (PRC), which describes the response of the $i^{th}$ oscillator to stimulation (i.e. speeding up or slowing down) depending on the precise timing of the stimulation with respect to the phase of $i^{th}$ oscillator, $\theta_i$. To simulate effective phase-locked stimulation, I was selected as a stimulation intensity value that resulted in desynchronization of population activity during continuous, 130 Hz high frequency stimulation. In clinical practice identification of an effective stimulation intensity value, or amplitude, is undertaken by systematic, stepwise evaluation of stimulation amplitudes from the clinical parameter space [43]. Neurons have been classified into two distinct categories based on their PRCs, that characterise their response to perturbation: type I PRCs either exclusively delay or advance spike firing, whereas type II PRCs can both advance or delay dependent upon the specific phase at which stimulation is delivered [44]. Cagnan et al. (2017) have demonstrated that neural oscillators in the context of pathological tremor exhibit type II PRCs [45]. Therefore, in this study, the PRC is described as $Z(\theta_i) = -\sin(\theta_i)$.

Under certain assumptions, the Kuramoto model can be solved to express the behaviour of each oscillator in terms of the order parameters, $\rho$ and $\psi$, describing the population's mean phase-coherence and mean phase, respectively.

$$\frac{d\theta_i}{dt} = \omega_i + \gamma\rho sin(\psi - \theta_i) + IX(t)Z(\theta_i) \tag{3}$$

The population's mean phase-coherence, $\rho$, represents synchrony where $\rho = 0$ and $\rho = 1$ correspond to either complete desynchrony or synchrony of the population, respectively. In

this manner, $\rho$ can be used as a control signal, as currently used in adaptive DBS to capture the instantaneous biomarker power, or level of synchronization, in specific frequency bands [37,46]. In line with clinical and experimental literature, we assume that lower levels of $\rho$ corresponds to reduced disease symptoms while higher levels of $\rho$ reflects increased symptom severity. The performance of the TV-BayesOpt algorithm can therefore be assessed by quantifying how the precise stimulation timing i.e., the optimal $\psi$ at which to apply stimulation, or intensity of stimulation can minimize population synchrony (in terms of $\rho$).

The objective function (i.e., the mapping between stimulation settings and symptom suppression) for the TV-BayesOpt algorithm is represented as the effect of phase-locked stimulation on neural synchrony ($\rho$). This can be quantified with an amplitude response curve (ARC) where the ARC represents the change in population synchrony ($\rho$) due to stimulation delivered at a particular phase ($\psi$). The ARC therefore represents the underlying objective function where certain phases of stimulation may increase or decrease population synchrony. The shape of a population ARC can be varied over time by updating the PRC ($Z(\theta_i)$) for the Kuramoto oscillators to emulate gradual or periodic changes.

## 2.3. Bayesian optimization

### 2.3.1 Formulation of the surrogate model—the Gaussian Process prior.
To estimate the form of the ARC that maps from stimulation parameters to population synchrony (i.e., the objective function), we implemented a surrogate model based on a Gaussian Process (GP) prior. The GP prior model can be thought of as a probability distribution over a set of non-linear regression functions (as opposed to a Gaussian distribution which is a set over variables). This permits efficient estimation of a mean function, as well as its relative uncertainty across the parameter space, indicating the confidence in the evaluated objective function for each parameter value.

The objective function $f(x)$ is fully defined by its mean $\mu(x)$ and covariance $K(x, x')$, that form sufficient statistics such that $f(x) \sim GP(\mu(x), K(x, x'))$, where $x$ in this context are the stimulation parameters to be explored. The covariance function describes the relationship between points in the parameter space, and is vital in determining the form (e.g., smoothness, or periodicity) of the estimated objective function. Thus $K_s(x, x')$ describes the *spatial* covariance (i.e., the space between parameter x and its neighbour x'). Our choice of covariance function is described later (see section "2.3.3 Spatial Covariance Function Selection").

The mean and covariance functions allow us to derive a joint distribution of the noisy outputs $y$ (previously sampled simulation parameters) and the estimated value of the objective function (mapping between stimulation settings and symptom suppression) $f^*$ evaluated at a new sample point $x^*$ in the parameter space. This is commonly defined as:

$$\begin{bmatrix} y \\ f_* \end{bmatrix} \sim \left( \begin{bmatrix} \mu(\boldsymbol{x}) \\ \mu(x_*) \end{bmatrix}, \begin{bmatrix} \boldsymbol{K}_s(\boldsymbol{x}, \boldsymbol{x}) + \sigma_n^2 \boldsymbol{I} & \boldsymbol{K}_s(\boldsymbol{x}, x_*) \\ \boldsymbol{K}_s(x_*, \boldsymbol{x}) & K_s(x_*, x_*) \end{bmatrix} \right) \tag{4}$$

where $\boldsymbol{y} \sim GP(\boldsymbol{\mu}(\boldsymbol{x}), \boldsymbol{K}_s(\boldsymbol{x}, \boldsymbol{x}) + \sigma_n^2 \boldsymbol{I})$, $\boldsymbol{x} = [x_1, \ldots, x_n]$ and $\boldsymbol{y} = [y_1, \ldots, y_n]$. As additional samples from the parameter space are evaluated, the prior is updated to form a posterior distribution to improve the model's approximation of the shape of the objective function, $f(x)$. Conjugacy between the prior and likelihood function (i.e. a function which estimates how probable it is to observe the outcome values $\boldsymbol{y}$ at the sampled values $\boldsymbol{x}$) allows the posterior to be computed analytically by omitting the evidence term (i.e. the integral of the likelihood function over the entire parameter space). The predictive distribution can therefore be written as

$$p(f_*|D_n, x_*) \sim N(\mu_*, \sigma_*^2) \tag{5}$$

Where $D_n = \{x, f\}$ represents the previously sampled parameter values and their respective outcome values and the posterior distribution is defined as

$$\mu_* = K_s(x_*, x)[K_s(x, x) + \sigma_n^2 I]^{-1} y \tag{6.a}$$

$$\sigma_*^2 = K_s(x_*, x_*) - K_s(x_*, x)[K_s(x, x) + \sigma_n^2 I]^{-1} K_s(x, x_*) \tag{6.b}$$

**2.3.2. Acquisition function.** The acquisition function is used to identify regions of the parameter space which should be tested during the optimization process. To determine the next sample to test during the optimization process, the acquisition function $\alpha(x)$ uses the current estimate of the underlying surrogate model and calculates the expected utility of all samples in the parameter space. The next sample to be tested is subsequently selected as the sample point $x^*$ which minimizes the acquisition function:

$$x_* = \underset{x_*}{\operatorname{argmin}} \ \alpha(x; D_n) \tag{7}$$

A Gaussian Process—Lower Confidence Bound (GP-LCB) acquisition function was chosen for the BayesOpt algorithm as it allows flexible tuning of parameter space exploration during the optimization process. The GP-LCB is defined as

$$\alpha_*(x_*) = \mu_*(x_*) - \kappa_n \sigma_*(x_*) \tag{8}$$

where the parameter $\kappa_n$ rescales $\alpha_*$ the uncertainty estimate of the GP. To balance between exploration of under sampled regions of the parameter space, or exploit already sampled regions, the value of $\kappa_n$ in the acquisition function is chosen according to the specific needs of the user where high $\kappa_n$ values favour exploration, and low $\kappa_n$ values favour exploitation of the parameter space. The role of $\kappa_n$ for guiding this exploration-exploitation trade-off when sampling from the parameter space between optimization steps is illustrated in Fig 1 in which we show an example of either maximally explorative or exploitative acquisition functions.

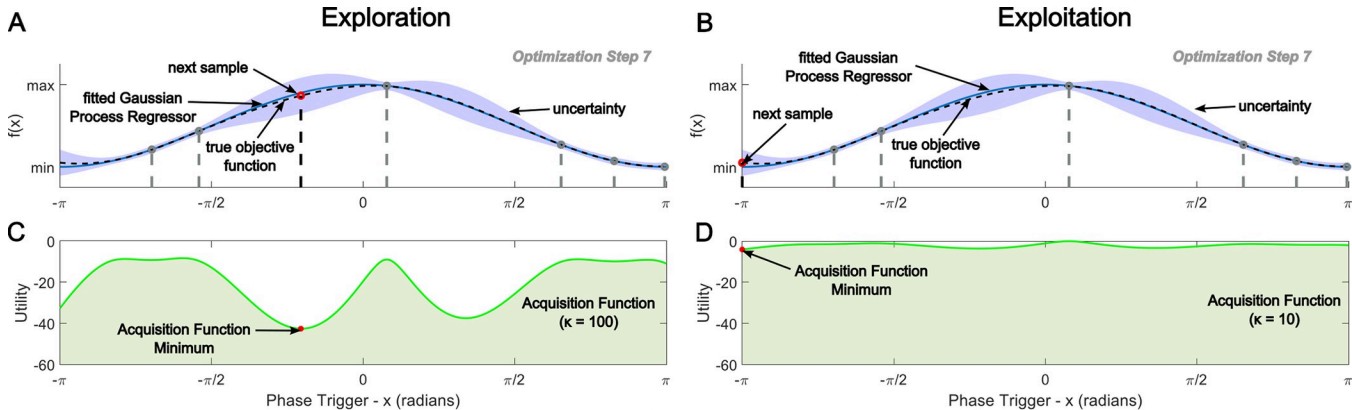

**Fig 1. Parameter space exploration in comparison to exploitation for the selection of samples during Bayesian Optimization.** Panels A and B illustrate the true (dotted black line) and estimated (solid blue line) shape of the underlying objective function. The confidence bounds for the estimated shape of the objective function are highlighted in pale blue while the associated samples previously taken and used for the estimation are represented as grey circles. Selection of the next parameter value to be tested (displayed as a red circle) is subsequently determined as the minimum of an acquisition function calculated from the mean and confidence bounds of the current estimated shape of the objective function. Panels C and D illustrate the corresponding lower confidence bound acquisition function, where the value k determines whether the algorithm prioritizes exploring regions of the parameter space where there is greater uncertainty (exploration) or prioritizes selecting regions of the parameter space where there is less uncertainty (exploitation).

**2.3.3. Spatio-temporal covariance functions.** To track a dynamically varying set of optimal therapy parameters, we eschewed the common assumption that the objective function is time-invariant. When applied to time-varying systems, the standard BayesOpt routine will exhibit diminished performance as it assumes all samples have equal contribution to the current estimate of the objective function. In contrast, a TV-BayesOpt algorithm weighs the contribution of previously tested samples to the current estimate based on the time at which samples were taken in relation to the current estimation step. Sources of temporal variation which lead to changes in the optimal parameter(s) may be gradual (e.g., linear), such as during disease progression, or due to other factors such as biological rhythms, like the circadian sleep-wake cycle or medication cycles (e.g., periodic). The goal of the TV-BayesOpt algorithm is thus to find the optimal parameter set for a given time, i.e., the parameters which provide the best suppression of symptoms, and to subsequently track the development of this parameter set over time.

To meet these requirements, a spatio-temporal covariance function was constructed. The spatio-temporal covariance function was comprised of two subcomponents which are referred to as the spatial and temporal covariance functions. The spatial covariance function in this context is equivalent to the covariance function used for time-invariant BayesOpt which captures the prior knowledge regarding the shape of the objective function. On the other hand, the temporal covariance function captures the anticipated variation in the shape of the objective function over time, i.e., rescaling the relevance of previously sampled data in relation to the most recent sample. The resulting spatio-temporal covariance function, $\tilde{K}(x, x')$, is formulated as the Hadamard product of the spatial, $K_s(x, x')$, and the temporal covariance functions, $K_t(t, t')$:

$$\tilde{K}(x, x') = K_s(x, x') \circ K_t(t, t') \tag{9}$$

A multitude of covariance functions exist across the field of machine learning that are capable of describing complex topologies [47]. The performance of the BayesOpt algorithm is thus dependent on the selection of an appropriately shaped covariance function for the problem that is being investigated.

## Spatial covariance function selection

Here, we aim to identify an optimal phase for phase-locked stimulation to suppress a rhythmic biomarker of disease. Therefore, the parameter space (target phase values) is periodic, where -π and π radians represent the same point in the parameter space. Therefore, the spatial covariance function (i.e., the covariance between target stimulation phases in the parameter space) was selected to also be periodic using the exponential sine squared covariance function:

$$K_s(x, x') = \exp\left(-\frac{2\sin^2\left(\frac{\pi|x-x'|}{T_x}\right)}{l_x^2}\right) \tag{10}$$

where hyperparameters $l_x$ and $T_x$ are the lengthscale and period (incorporating our prior knowledge of the periodicity of the parameter space).

## Temporal covariance function selection

For biological systems, there are several sources of variations in the shape of the objective function over time, including disease progression (leading to monotonic changes) and biological ryhthms such as the circadian sleep-wake cycle (leading to periodic changes), or a combination of the two. To overcome these temporal variations, we outline the selection of appropriate

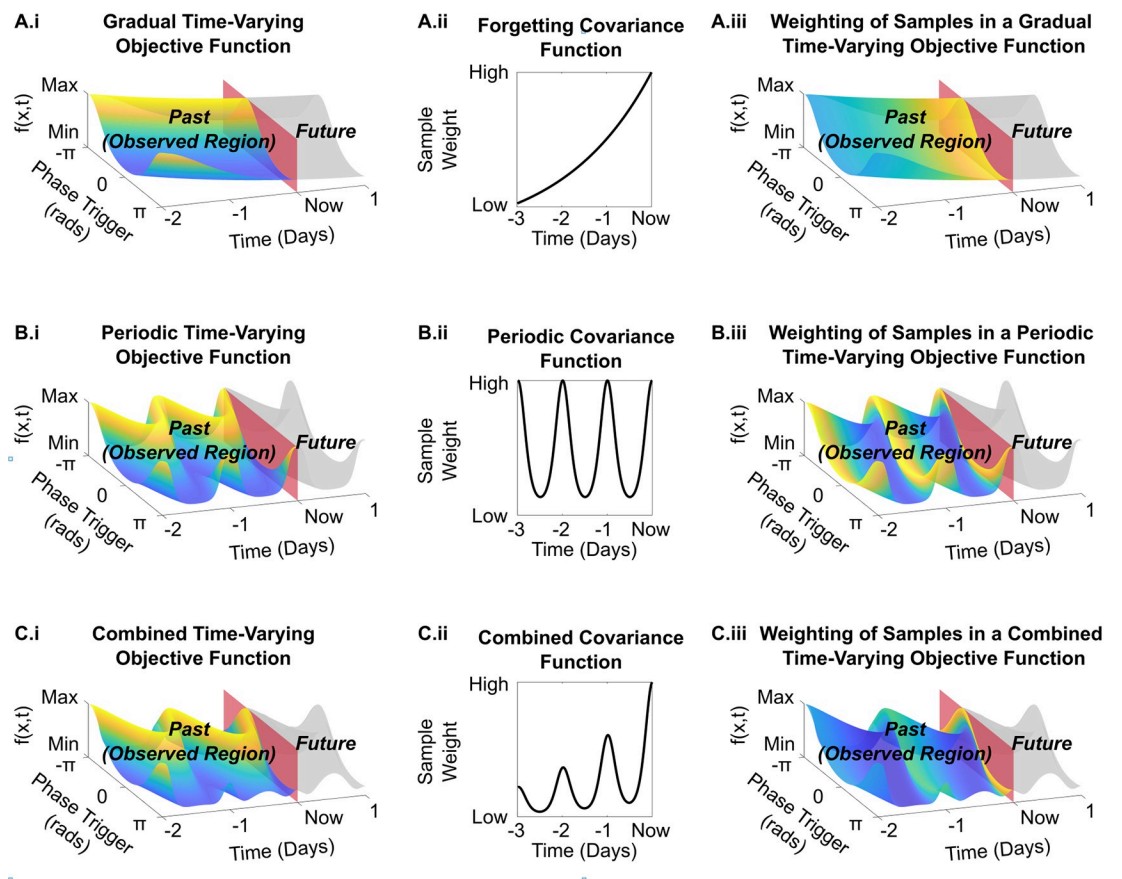

**Fig 2. Time-Varying objective functions and their respective covariance functions.** The various drifts explored in this section are illustrated for a periodic objective function with respect to the parameter of interest (stimulation phase). This objective function is allowed to drift according to three cases: gradual (A.i), periodic (B.i), and the superposition of the two (C.i). At each time-step in the optimization, samples from the objective function are weighed by their respective temporal covariance function (depicted in panels A-C. ii). This resultant weighting is illustrated in panels A-C.iii in the colour gradient of the drifting objective function, with brighter colours representing a stronger sample weight.

temporal covariance functions in the following sections, summarizing the topology of these covariance functions in Fig 2.

## Temporal covariance function–gradual "Forgetting"

To accommodate gradual variations in the shape of the objective function a 'forgetting' covariance function that reduces the influence of older samples on the current estimate can be implemented, Fig 2A (i-iii). Bogunovic et al. (2016) introduced a covariance function [48], defined as

$$K_t(t, t') = (1 - \epsilon)^{\frac{|t-t'|}{2}} \tag{11}$$

where $t$ is the current sample time, $t'$ is the time at which previous samples were taken and $\epsilon$ is referred to as the forgetting factor. The forgetting factor, $\epsilon$, has a value between [0, 1) which determines how quickly the contribution of previously taken samples to the current estimate are reduced. A forgetting factor $\epsilon = 0$ corresponds to no data being forgotten and all previous samples contributing equally to the estimate of the objective function, while $\epsilon$ values close to 1 correspond to previous data samples being forgotten so fast that only the current sample is used to estimate the objective function.

An appropriate value of $\epsilon$ can be learned using past data through maximum likelihood estimation or empirically selecting $\epsilon$ to reduce the contribution of previous samples based on a specified half-life:

$$\epsilon = \lambda = \frac{\ln 2}{t_{1/2}} \tag{12}$$

where $\lambda$ corresponds to the half-life decay factor and $t_{1/2}$ defines half-life, specified as number of samples, i.e. a $t_{1/2}$ value of 100 corresponds to the objective function value associated with the 100<sup>th</sup> last sample taken contributing only 50% of its orignal value to the current estimate of the surrogate model.

## Temporal covariance function–periodic

To accommodate repeating variations in the shape of the objective function, a periodic covariance function can be implemented to weight the contribution of previously sampled data based on specific time points in the cycle at which the samples were taken, Fig 2B (i-iii). In this manner, the contribution of samples taken at time points near the same parts of the cycle are weighted more similarly to those taken at other time points in the cycle which are weighted less. This behaviour can be captured similar to the spatial covariance function by using the exponential sine squared covariance function.

$$K_t(t, t') = \exp\left(-\frac{2\sin^2\left(\frac{\pi|t-t'|}{T_t}\right)}{l_{t,p}^2}\right) \tag{13}$$

where $T_t$ is the temporal period of the underlying biomarker oscillation (defined with the same units as $t$) and $l_{t,p}$, the length-scale, controlling the amplitude of the temporal oscillation. In practice, the units of $T_t$ will depend on the sampling protocol and may be defined in either sample counts or physical timescales such as hours or minutes. Furthermore, $T_t$ can be estimated empirically by tracking the variation in symptom suppression under fixed stimulation parameters.

## Temporal forgetting-periodic covariance function

For implementations in real-life scenarios where both gradual and periodic variations may be superimposed, the forgetting and periodic temporal covariance functions can be combined, Fig 2C (i-iii). Doing so produces the temporal covariance function below which is the product of the forgetting covariance function and the periodic covariance function.

$$K_t(t, t') = (1 - \epsilon)^{\frac{|t-t'|}{2}} \exp\left(-\frac{2\sin^2\left(\frac{\pi|t-t'|}{T_t}\right)}{l_{t,p}^2}\right) \tag{14}$$

where $\epsilon$, $l_{t,p}$ and $T_t$ are the covariance function hyperparameters as described above in the previous sections. This temporal covariance function is used to reduce the contribution of older samples on the current estimate while also flexibly tracking periodic variations in the shape of the objective function over time, as depicted in Fig 2C (ii).

## 2.4. Simulation details

The Kuramoto model of DBS was initially simulated for a 200 s period to allow the model to reach its steady state behaviour. The initial phase of each oscillator in the population was selected from a uniform distribution between $[0, 2\pi]$ at the start of the simulation. After 200 s,

the model was subsequently simulated for 3000 optimization steps where each optimization step corresponded to a 58 s simulation time. During each optimization step, phase-locked stimulation at a target phase value was applied only during the last 8 s of the 58 s simulation period. The initial 50 s of each optimization step was included to allow the model to return to its steady state following application of phase-locked stimulation in the previous optimization step. The effectiveness of DBS after each optimization step, $y$, was quantified as the normalized change in the mean phase-coherence of the population during the stimulation period from its baseline value:

$$y = \Delta\rho = \frac{\overline{\rho_{stim}} - \overline{\rho_{baseline}}}{\overline{\rho_{baseline}}} \tag{15}$$

where $\overline{\rho_{stim}}$ is the phase-coherence of the population averaged over the 8 s stimulation period and $\overline{\rho_{baseline}}$ is the baseline population phase-coherence, calculated over the 25 s period prior to stimulation application.

To initiate the TV-BayesOpt algorithm, data from the parameter space was needed. To this end, 12 equally spaced samples between [-π, π] were tested to coarsely characterize the population response to phase-locked stimulation at different stimulation phase values. Following this, selection of subsequent phase values was determined by the TV-BayesOpt algorithm as described above.

Drifts in the objective function (mapping between stimulation settings and symptom suppression) were simulated by incrementally adding a phase offset value, $\Delta\theta$, to the population PRC: i.e., $Z(\theta_j) = -\sin(\theta_j + \Delta\theta)$. This corresponded to the location of the optimal stimulation phase, $\psi^*$, being incrementally varied over the simulation duration. This is illustrated in Fig 3 where the change in population ARC, which represents the change in population synchrony due to stimulation delivered at a particular phase ($\psi$), and PRC are highlighted.

At each optimization step, the associated regret of the algorithm was calculated as

$$R = f(\psi_k^*) - f(\psi_{target_k}) \tag{16}$$

Where $\psi_k^*$ is the location of the true optimum value for phase-locked stimulation at optimization step k, $\psi_{target_k}$ is the stimulation phase value tested at the kth optimization step, $f(^*)$ is the measured objective function value in response to stimulation at the specified input phase value, i.e. the change in the population mean-phase coherence due to stimulation at the specified input target phase value. Thus, at each optimization step the regret quantified the difference between the suppression achieved by the algorithm in comparison to the maximum possible suppression that was achievable at that optimization step. In a practical implementation this would correspond to the difference between the symptom suppression achieved by the algorithm, objectively quantified based on an accelerometer or LFP power signal at the frequency band of interest recorded from a patient, and the maximum symptom suppression that was possible at that optimization step. The overall performance of the algorithm at each step over the optimization process was subsequently quantified using the cumulative regret, defined as

$$CR = \frac{\sum_{k=1}^{n}(f(\psi_k^*) - f(\psi_{target_k}))}{n} \tag{17}$$

where $n$ is the total number of optimization steps. The cumulative regret thus quantifies the total regret that is accumulated over the entire optimization process.

A gradual drift was simulated as a phase offset advancing from 0 to—π over 3000 optimization steps, a periodic drift was simulated as a sinusoidal phase offset advancing from 0 to–π

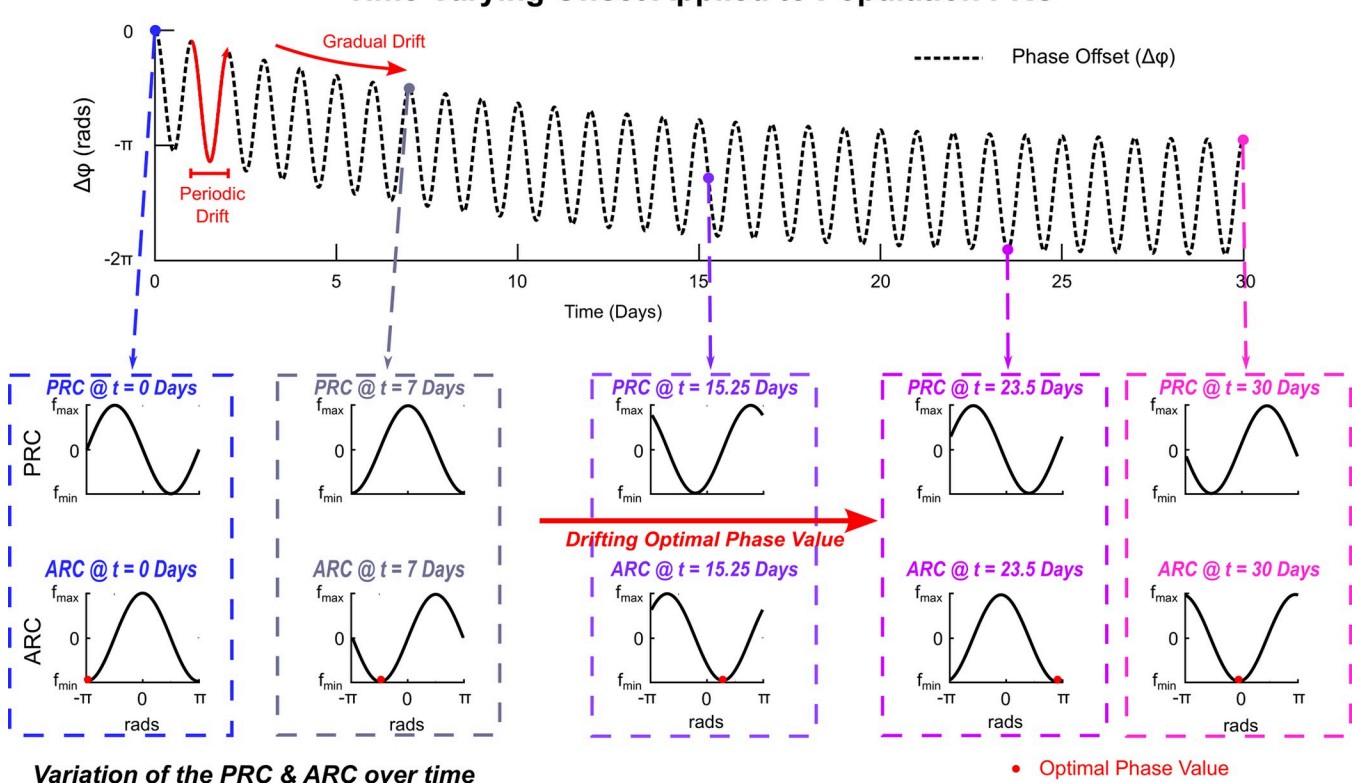

**Fig 3. Illustration of Dynamic Variation of the population ARC and PRC.** Illustration of the shape of the initial population ARC corresponded to a PRC of $Z(\theta_i) = -sin(\theta_j)$ *at the first optimization step (t = 1).* By incrementally adding an offset value, $\Delta\theta$, to the PRC, the shape of the ARC and location of the optimum phase value for phase-locked stimulation, $\psi^*$, were gradually varied over the optimization process. For each PRC and ARC $f_{max}$ and $f_{min}$ correspond to the phase trigger values that produced the maximum and minimum attenuation of the synchronous activity in the KM model, respectively.

and returning back to 0 to over 100 optimization steps, and lastly the superimposed gradual and periodic drifts were simulated as the superposition of the gradual and periodic drifts. For each simulated scenario, the hyperparameters of the utilized temporal covariance functions were independently selected based on the anticipated temporal drift in the data. In this manner, simulations of only gradual or periodic drifts in the phase offset utilized gradual "forgetting" or periodic temporal covariance functions in the algorithm, respectively. Meanwhile in the superimposed drift scenario, the hyperparameters for the gradual "forgetting" and periodic temporal covariance functions were respectively selected based on these anticipated components of the overall simulated drift. For a real-world implementation, longitudinal recordings of biomarker signal(s) (derived from an accelerometer or electrophysiological recordings) should be undertaken prior to algorithm deployment to characterize patient-specific temporal variations in therapy (potentially due to the 24-hour circadian rhythm or patient medication cycles). These characterized temporal variations can be subsequently incorporated as a priori information for the algorithm when selecting appropriate hyperparameter values, i.e. the periodic covariance function period.

Finally, sensitivity analyses were conducted to investigate the influence of the temporal covariance forgetting factor, $\epsilon$, and period, $T_t$, on the TV-BayesOpt algorithm's performance at tracking a gradual and periodic temporal drift and purely periodic temporal drift, respectively.

Model simulation, post-processing and signal analysis were conducted using custom scripts developed in MATLAB (MathWorks, Inc., Natick, MA). The source code used to produce the

**Table 1. Summary of Kuramoto DBS Model Parameters.**

| Model Parameter | Parameter Description | Value |
|:---:|:---|:---:|
| $N$ | Number of oscillators in population | 50 |
| $f_0$ | Natural frequency of the population oscillators | 8 Hz |
| $\omega$ | Natural frequency of the population oscillators | $2\pi * f_0$ |
| $\sigma$ | Standard deviation of population oscillators from the natural frequency, $\omega_{biomarker}$ | 0.0075 |
| $\gamma$ | Coupling strength between the oscillators in the population | 0.8 |
| $I$ | Stimulation intensity | 30 |

results and analyses presented in this manuscript is available for download from the OUI Software Store (https://process.innovation.ox.ac.uk/software/) with reference 21242. Parameter values for the Kuramoto model in (2) are included below in Table 1.

# 3. Results

This study develops and evaluates the performance of a time-varying optimizer to determine the optimum stimulation parameter set. We test the algorithm's performance in the context of an oscillopathy, Essential Tremor, in order to determine the optimum stimulation phase when this value would change gradually as a result of disease progression and/or periodically due to patient's sleep/wake cycle. However, these observations can be extended to any other neurological and psychiatric condition, which is driven by rhythmic neural activity, and other stimulation parameters (e.g., amplitude, pulse width and frequency).

## 3.1. Kuramoto model behaviour

We implemented a coupled Kuramoto neural oscillator model to test the efficacy of the TV-BayesOpt algorithm in tracking gradual and/or periodic changes in the optimum stimulation phase. The model reached its steady state with an order parameter $\rho$ (i.e., population synchrony taken to be a direct correlate of symptom severity) of 0.8, reflecting a highly synchronized state in the absence of stimulation after a 200 s transient period. The population's response to phase-locked stimulation was investigated using a static PRC, set to $-\sin(\theta)$. This represents a type 2 oscillator, in line with previous experimental observations from phase-locked deep brain stimulation in Essential Tremor [45]. Stimulation was initially applied at 12 equally spaced population phase values, $\psi_{target}$, ranging between $(-\pi,\pi)$ (Fig 4). Phase-locked stimulation at $\psi_{target} = \left(-\pi, -\frac{\pi}{2}\right)$ radians and $\psi_{target} = \left(\frac{\pi}{2}, \pi\right)$ radians resulted in desynchronization of the population and a reduction in mean phase-coherence ($\Delta\rho<0$), in line with theoretical expectations. In contrary, phase-locked stimulation applied between $\psi_{target} = \left(-\frac{\pi}{2}, \frac{\pi}{2}\right)$ radians resulted in increased population synchrony which was reflected in an increase in the population mean phase-coherence ($\Delta\rho>0$). In oscillopathies such as Parkinson's disease and Essential Tremor, stimulation is used to reduce exasperated neural synchrony, which is correlated with patients' symptom severity. Therefore, the "optimal phase value" for stimulation, $\psi^*$, was identified as the value that resulted in the greatest reduction in population synchrony. This value was identified as $\psi^* = -\pi$ radians. After this initial "burn-in" period, the location of $\psi^*$ was dynamically varied as described in the subsequent sections.

## 3.2. Algorithm performance

**3.2.1. Gradual drift.** A gradual drift in the optimal stimulation setting may be anticipated due to disease progression in common neurological and psychiatric conditions. Such a change

## 1) Simulate Model to Steady State

## 2) Apply Phase-locked Stimulation at different Phase Values (x rads)

## 3) Fit Surrogate Model based on Initial Trials

**Fig 4. Kuramoto DBS model response to phase-locked stimulation.** Individual oscillators in the population are represented as blue circles distributed around the unit circle where the position of each oscillator represents its phase value at the associated timestep. The population mean phase-coherence at different simulation timesteps is illustrated in 2). In the top row of 2) the red line within the unit circle illustrates the magnitude of the population mean phase-coherence, $\rho$, and the mean phase of the population, $\psi$. The red line points in the direction of $\psi$ and the length of the line represents the synchronization level of the population $\rho$ (a short line represents low synchrony, and a long line represents high synchrony). The 1) panel represents the distribution of oscillators prior to stimulation when the model is at its steady state behaviour. In panel 2) the rows illustrate the change in population synchrony in response to stimulation at a specific phase value, $\psi_{target}$. The bottom row summarizes the change in population synchrony, $\Delta\rho$, as a function of phase-locked stimulation applied between $\psi_{target} = (-\pi, \pi)$ radians. $\Delta\rho < 0$ corresponded to phase-locked stimulation at the specified $\psi_{target}$ values desynchronizing the population, while $\Delta\rho > 0$ corresponded to the phase-locked stimulation increasing population synchrony.

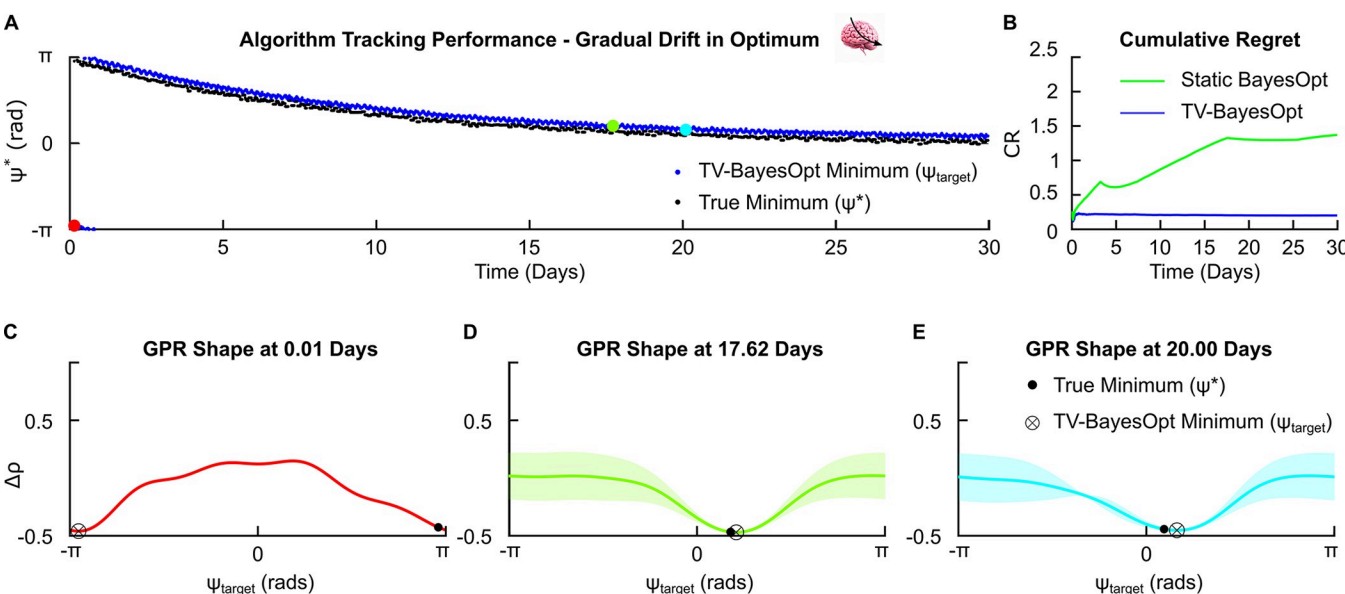

**Fig 5. TV-BayesOpt algorithm performance for tracking a gradual drift in the optimal stimulation phase for phase-locked stimulation, $\psi^*$.** Panel A illustrates the tracking performance of the TV-BayesOpt algorithm (blue dots) at locating the true optimal phase value (black dots) for population desynchronization. Panel B illustrates the associated average regret for the TV-BayesOpt algorithm at tracking the true optimum phase value in comparison to when static BayesOpt was implemented alone. Panels C-E illustrate the true location of the optimal stimulation phase (black dot), the minimum predicted by the TV-BayesOpt algorithm (black circle and cross) and the shape of the GPR predicted by the TV-BayesOpt algorithm at 0.01, 17.62 and 20.00 day time points. For each estimated GPR the confidence bounds observed at the predicted optimal phase value are small and become larger for values further away from this value due to the algorithm's acquisition function prioritizing exploitation of the parameter space during the optimization process.

in the stimulation phase, $\psi^*$, ranging from π to 0 radians over 30 days was continuously tracked using the TV-BayesOpt algorithm (Fig 5A). The TV-BayesOpt algorithm with the "forgetting" temporal covariance function $K_t(t, t') = (1 - \epsilon)^{\frac{|t-t'|}{2}}$ (see section 2.3 "Temporal Covariance Function–'Gradual Forgetting'") resulted in lower cumulative regret over the optimization process, calculated as the area under the curve (AUC) of the cumulative regret plot, when compared to the time-invariant implementation of the BayesOpt algorithm (Fig 5B). Here, cumulative regret represents the difference between the true optimal stimulation phase and the one identified by the optimization algorithm. ± π offset between true optimal and predicted phase values would result in population synchrony not being different from baseline untreated levels. Under the assumption that synchrony reflects symptom severity, this would be equivalent to symptom severity not changing. The shape of the TV-BayesOpt estimated Gaussian Process Regressor (GPR) at different timepoints is displayed in Fig 5C–5E. At each timepoint the estimated GPR had low uncertainty in the region of the parameter space around $\psi^*$ while this uncertainty is gradually increased for regions more distant from $\psi^*$.

**3.2.2. Periodic drift.** A periodic drift, characterizing changes in the effective stimulation parameters due to circadian cycles, was emulated by varying $\psi^*$ from π to 0 radians over 100 optimization steps, representing a 1-day period. Implementation of the TV-BayesOpt algorithm with a temporal periodic covariance function $K_t(t, t') = \exp\left(-\frac{2\sin^2\left(\frac{\pi|t-t'|}{T_t}\right)}{l_{t,p}^2}\right)$ (where $T_t = 100$; see section 2.3 "Temporal Covariance Function–Periodic") led to accurate tracking of the location of $\psi^*$ (Fig 6A). The temporal periodic covariance function with a one-day period led to better tracking of $\psi^*$ in comparison to static BayesOpt, as evidenced by a lower cumulative regret AUC (Fig 6B). Beyond circadian cycles, such a periodic change could also be expected

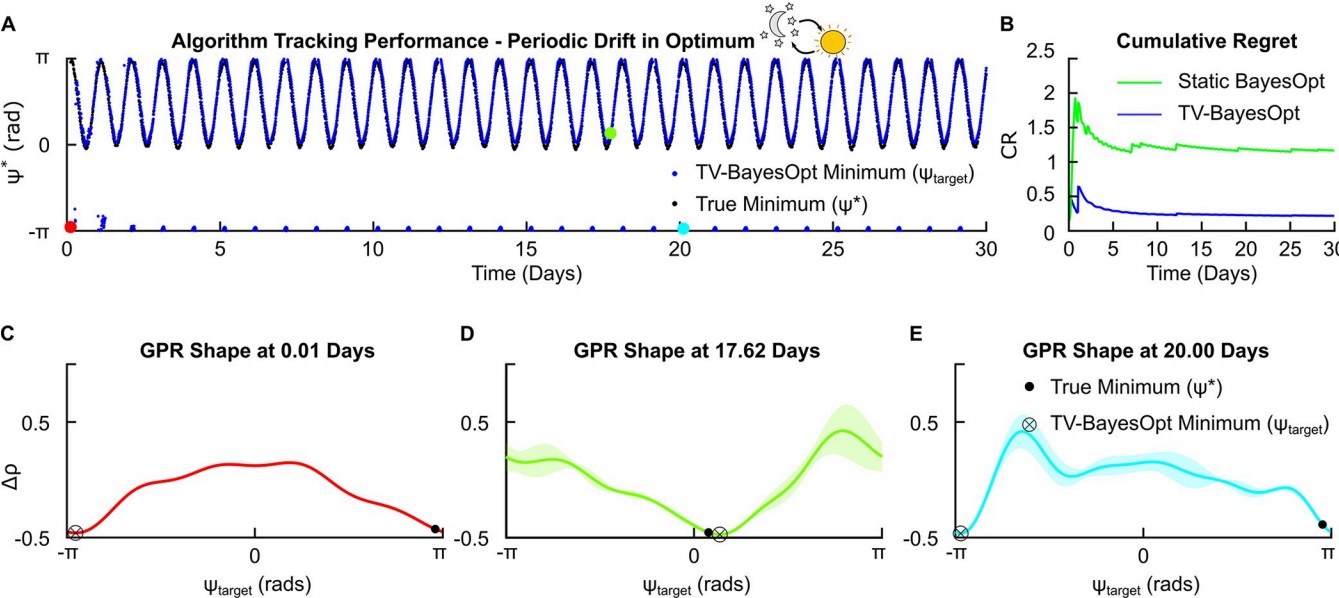

**Fig 6. TV-BayesOpt algorithm performance for tracking a periodic drift in the optimal stimulation phase for phase-locked stimulation, $\psi^*$.** Panel A illustrates the tracking performance of the TV-BayesOpt algorithm (blue dots) at locating the true optimal phase value (black dots) for population desynchronization. Panel B illustrates the associated average regret for the TV-BayesOpt algorithm at tracking the true optimum phase value in comparison to when static BayesOpt was implemented alone. Panels C-E in the bottom row illustrate the true location of the optimal stimulation phase (black dot), the minimum predicted by the TV-BayesOpt algorithm (black circle and cross) and the shape of the GPR predicted by the TV-BayesOpt algorithm at 0.01, 17.62 and 20.00 day time points. For each estimated GPR the confidence bounds observed at the predicted optimal phase value are small and become larger for values further away from this value due to the algorithm's acquisition function prioritizing exploitation of the parameter space during the optimization process.

from medication intake, and other repeating patterns which may influence therapy efficacy. To account for other rhythms, the $T_t$ should be altered to reflect these changes (e.g., 3 hours for medication intake, etc).

**3.2.3. Superimposed drift.**   We next explored the performance of our algorithm in the presence of both gradual and periodic changes in the optimal stimulation settings. Superposition of gradual and periodic drifts required the use of the temporal forgetting-periodic covariance function $K_t(t, t') = (1 - \epsilon)^{\frac{|t-t'|}{2}} \exp\left(-\frac{2 \sin^2\left(\frac{\pi|t-t'|}{T_t}\right)}{l_{t,p}^2}\right)$ (see section 2.3 "Temporal Forgetting-Periodic Covariance Function") and produced a lower cumulative regret AUC in comparison to the time-invariant BayesOpt algorithm (Fig 7A). This illustrates the "composite" nature of the algorithm, whereby the temporal covariance function can be combined to account for different scenarios in a disease process.

### 3.3 Algorithm performance sensitivity analysis

**3.3.1. Forgetting factor.**   One of the critical factors which influences the algorithm performance is the forgetting factor, $\epsilon$. To evaluate this interaction, the AUC of the cumulative regret plot was calculated for tracking a combination of gradual and periodic drifts, with high values indicating inaccurate optimization (Fig 8). We also contrasted the algorithm performance when solely a "forgetting" covariance function was used versus a "forgetting-periodic" covariance function (Fig 8) in order to evaluate the impact of selecting a "suboptimal" temporal covariance function i.e. when a priori knowledge regarding variations in the optimal stimulation parameter set is considered (forgetting-periodic covariance function) vs not (forgetting covariance function). For $\epsilon$ values less than 0.04, the forgetting covariance function resulted in

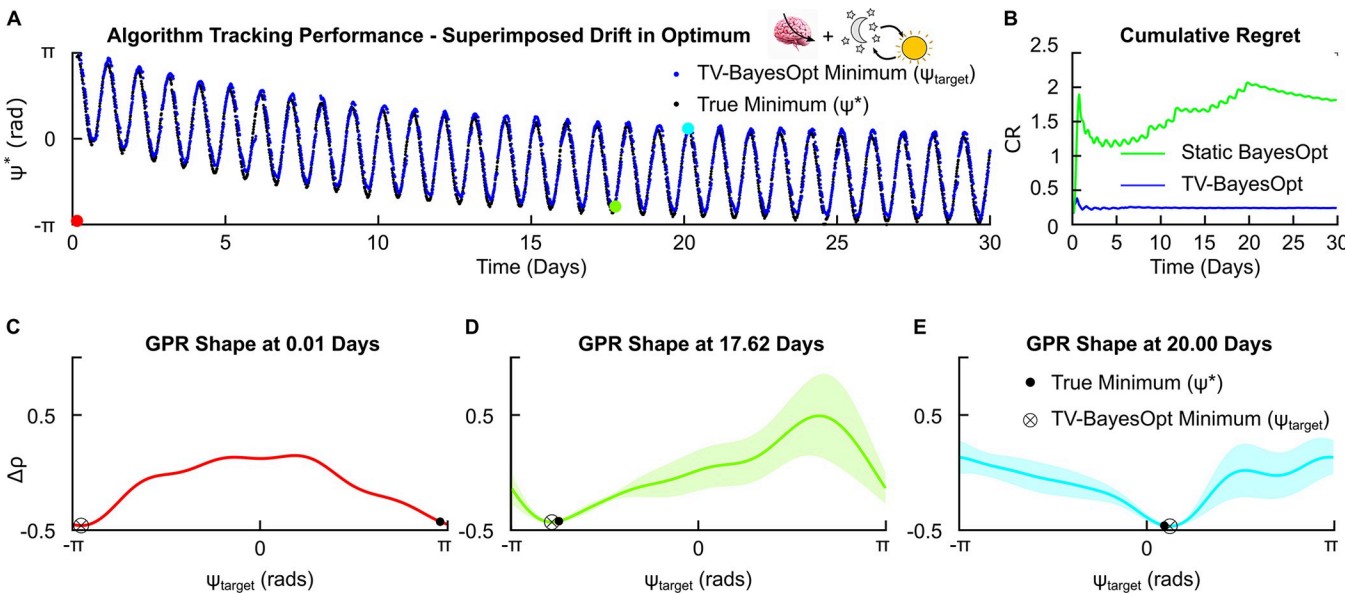

**Fig 7. TV-BayesOpt algorithm performance for tracking a superimposed (gradual and periodic) drift in the optimal stimulation phase for phase-locked stimulation, $\psi^*$.** Panel A illustrates the tracking performance of the TV-BayesOpt algorithm (blue dots) at locating the true optimal phase value (black dots) for population desynchronization. Panel B illustrates the associated average regret for the TV-BayesOpt algorithm at tracking the true optimum phase value in comparison to when static BayesOpt was implemented alone. Panels C-E in the bottom row illustrate the true location of the optimal stimulation phase (black dot), the minimum predicted by the TV-BayesOpt algorithm (black circle and cross) and the shape of the GPR predicted by the TV-BayesOpt algorithm at 0.01, 17.62 and 20.00 day time points. For each estimated GPR the confidence bounds observed at the predicted optimal phase value are small and become larger for values further away from this value due to the algorithm's acquisition function prioritizing exploitation of the parameter space during the optimization process.

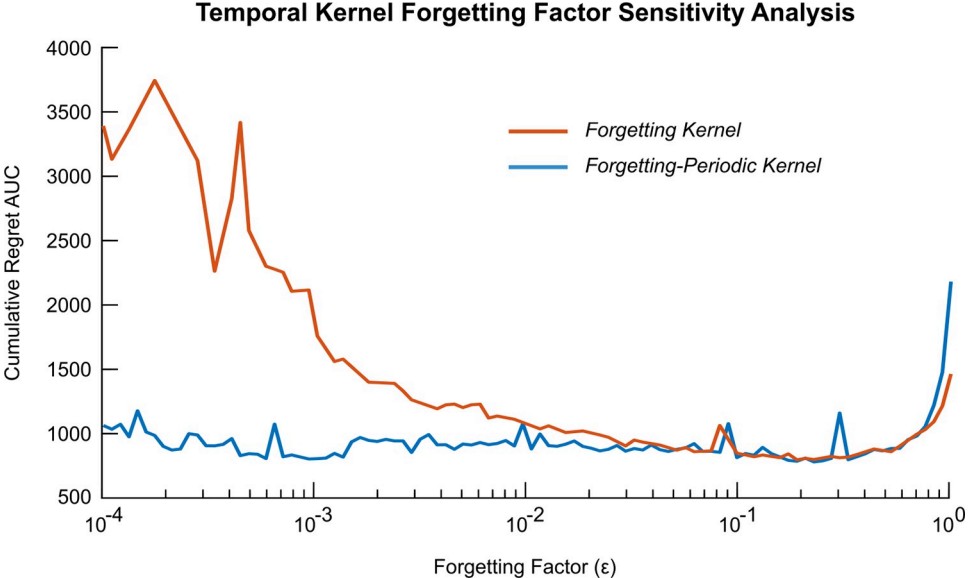

**Fig 8. Sensitivity analysis of the TV-BayesOpt algorithm with a forgetting (orange line) or a forgetting-periodic (blue line) covariance function for a range of $\varepsilon$ values.** Incorporation of prior knowledge of the temporal variation in the objective function optimum value (blue line) resulted in improved TV-BayesOpt algorithm performance than implementing a forgetting covariance function (orange line) alone. The performance of the algorithm at each forgetting factor value was calculated as the cumulative regret AUC value from the resulting cumulative regret plot generated at each respective forgetting factor value. Best algorithm performance was observed for an $\varepsilon$ value of 0.22, above this value the algorithm begins to forget previous samples too quickly to accurately track the optimal value in the objective function.

greater cumulative regret than the forgetting-periodic covariance function. Incorporating prior knowledge of periodicity led to a more stable performance of the algorithm. For $\epsilon$ values above 0.04, both covariance functions resulted in equivalent cumulative regret due to an appropriate alignment between the forgetting factor (i.e., the rate of forgetting of older data) and the simulated temporal drift in the optimum phase value. The algorithm provided the best performance with an $\epsilon$ value of 0.22 and corresponded to a data half-life ($t_{1/2}$) of 3.15 samples (representing a 45-minute simulation period). $\epsilon$ values above 0.22 resulted in increased cumulative regret AUC values as $\epsilon$ values in this range led to the algorithm forgetting data too quickly to accurately estimate the location of the optimal value.

**3.3.2 Variation in temporal drift period.** We next evaluated the algorithm's performance when the temporal drift anticipated by the algorithm, $T_t$, differed from the true rate of change in the optimal stimulation settings. The performance of the algorithm was also contrasted against a scheduler which made fixed adjustments to maintain stimulation at a certain value over a 24-hour period (Fig 9). The scheduler and TV-BayesOpt with a periodic covariance function were unable to track the location of the optimum stimulation phase when the drift in the optimal stimulation phase was different from the anticipated period, based on a priori information. This resulted in large AUC values for both approaches. When the forgetting-periodic covariance function was implemented, the algorithm was able to accommodate the difference between the true and anticipated temporal periods, as evidenced by the lower regret AUC value. The forgetting-periodic covariance function thus enabled the algorithm to more flexibly locate the optimum by limiting the influence of samples that were previously taken based on incorrect a priori information (i.e. an incorrect anticipated periodicity). With this, the location of the optimum could still be determined based on the most recently taken samples.

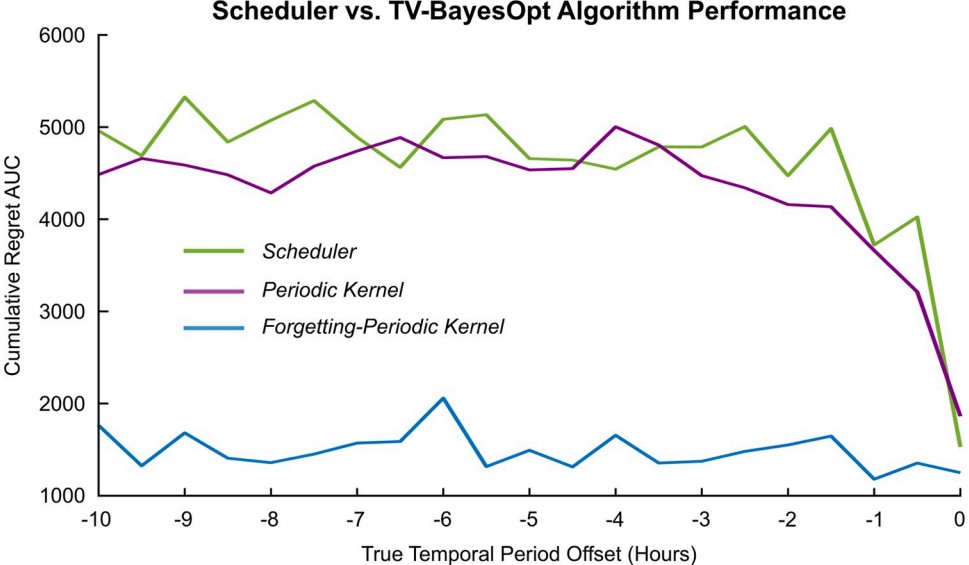

**Fig 9. Sensitivity analysis of the TV-BayesOpt algorithm performance at tracking a periodic temporal drift when the period of the drift is offset from the covariance function period anticipated by the TV-BayesOpt algorithm.** The performance of a scheduler (green line), the TV-BayesOpt with a periodic temporal covariance function (purple line) and the TV-BayesOpt with a periodic and smooth forgetting covariance algorithm were estimated by calculating the AUC value for the associated cumulative regret plot for each implementation at different offset in the true temporal period.

## 4. Discussion

Here, we introduce a new approach to therapy optimization that takes into account slow variations in therapy efficacy over time due to factors such as medication intake, circadian rhythms and disease progression. We explored the efficacy of the proposed approach in the Kuramoto model which is a well-established mathematical model that has been extensively used in literature to investigate synchronization in biological systems. The model has been previously used to simulate time variability in systems using time-dependent model parameters [49,50], in addition to modelling the effects of electrical stimulation on neural synchrony [42]. In this current study, slow variations in therapy efficacy were emulated by incremental changes in the shape of the model population PRC. By learning these variations, our time-varying controller sustains therapy efficacy in contrast to time-invariant approaches which show degraded performance.

In neurodegenerative disorders such as Parkinson's disease and Essential Tremor, therapy efficacy can be lost over time due to disease progression [51,52]. To counteract this, either the medication dose is increased or for those patients treated with DBS the stimulation amplitude is increased to capture a larger portion of the target nucleus. Other variations which influence stimulation efficacy include concurrent medication intake, which in a subset of Parkinson's disease patients may induce dyskinesias when used on its own or in conjunction with DBS. Beyond disease progression and medication intake, symptom variations over the course of the day, which may reflect circadian dependencies (i.e., Parkinson's disease, Essential Tremor, etc), could also influence therapy efficacy [12,27].

Brain stimulation-based therapies currently take a time-invariant approach to stimulation parametrisation. Optimal stimulation settings are chosen during day-time clinical visits through a process of trial and error and these settings remain the same until the next clinic visit. Adaptive brain stimulation approaches which switch ON stimulation when symptom severity or a representative neural biomarker exceeds a threshold, assume that the subject specific thresholds do not change over time. Other adaptive stimulation approaches such as phase-specific stimulation also do not take into account dynamic variations in effective stimulation settings and use the same stimulation phase. However, taking a time-invariant approach to stimulation parametrisation may either over stimulate or under stimulate patients, resulting in stimulation induced side-effects and sub-optimal symptom management, respectively.

### 4.1. Adaptive bayesian optimization algorithms

Previous literature has suggested a variety of algorithms for solving dynamic optimization problems [48,53–55]. In the context of Bayesian Optimization, many time-varying versions of the algorithm build on the adaptive Bayesian Optimization (ABO) algorithm presented in [48] which introduced the concept of smooth forgetting of older data samples by using a forgetting temporal covariance function. Variations of the ABO algorithm have been subsequently investigated by Imamura et al. (2020) who explored the ABO algorithm performance when the time between data samples was variable [56] and Chen & Li (2021) who implemented a threshold on sample freshness to discard old data when sudden system changes occur [57]. In contrast to using a smooth forgetting covariance function Nyikosa et al. (2018) explored an alternative strategy using stationary temporal covariance matrices whose length-scales were optimized from training data to determine the feasible prediction horizon for the time-varying algorithm [58].

In our current work we build on both Bogunovic and Nyikosa's algorithms to construct a TV-BayesOpt algorithm capable of tracking periodic variations in optimal stimulation parameters while also smoothly forgetting older data samples. The temporal covariance in our

proposed approach is not stationary due to our implementation of a periodic temporal covariance function. This means that covariance between two sampled points in time is no longer dependent solely on the distance between them, but rather on the absolute time at which the samples were taken. Data points which were sampled at the same phase of a repeating periodic rhythm, i.e., the 24-hour circadian rhythm, will have high covariance, while points taken at opposite phases of the rhythm will have a small covariance. In this manner, we leverage the advantages of both versions of the algorithm to enable smooth tracking of rhythmic variations in the optimal value which may be changing over time. From a neuromodulation perspective, this enables the proposed algorithm to track gradual and periodic variations in optimal stimulation parameters which may occur due to disease progression and biological rhythms, respectively. Moreover, incorporation of the smooth forgetting covariance function with a threshold on sample freshness can be used to define a limit on the number of data samples required for good algorithm performance in practical implementations of the algorithm as in [57]. Likewise, in the absence of gradual variations in optimal parameters, the algorithm may be used to characterize optimal stimulation parameters for the entire cycle of a periodic variation. In this manner, the algorithm could be hypothetically replaced with a scheduler that increments through optimal stimulation parameters over the course of the periodic variation (see section 3.4 "Algorithm Performance Sensitivity Analysis"). Synchronized stimulation adjustments in this manner have been previously proposed by Fleming et al. (2022) as an approach to optimize neuromodulation therapy in the presence of biological rhythms [12].

## 4.2. Implementation and utility

In this work, we highlight how implementation of the TV-BayesOpt algorithm utilizing both a smooth forgetting and periodic temporal covariance function leads to robust algorithm performance for tracking slow periodic variations (Figs 8 and 9).

Good algorithm performance may be achieved with smooth forgetting alone, however this requires careful selection of an appropriate forgetting factor well-matched to the temporal oscillation being tracked (Fig 8). When $\epsilon$ is too low, the algorithm weighs the contribution of older samples too heavily which leads to poor tracking of the optimum stimulation phase over time (Fig 8). When the TV-BayesOpt algorithm incorporates a periodic temporal covariance function whose period is aligned to the expected temporal drift in optimal stimulation settings, the performance is greatly improved for low $\epsilon$ values. Robust algorithm performance is subsequently observed for all $\epsilon$ values below 0.4. Above this value the algorithm performance begins to worsen due to previously taken data samples being forgotten too quickly. Thus, once periodic components of the temporal drift have been suitably characterized the practical trade-off of the algorithm achieving good performance is dependent on identification of an appropriate $\epsilon$ value. Selection of an $\epsilon$ value lower than necessary will result in the algorithm utilizing more, potentially redundant data as part of its inference, while too high a value may lead to an insufficient amount of available data for algorithm inference. Selection of an $\epsilon$ value thus requires a balance of these two considerations.

When there is no difference between the actual and expected variations in optimal stimulation parameters (Fig 9), the algorithm performs well with a periodic covariance function alone and maintains tracking of the optimal stimulation phase. In this case, once the algorithm has locked to the location of the optimum stimulation phase, its performance is equivalent to a scheduler that updates the location of the optimum stimulation phase value based on the anticipated temporal period (Fig 9). Worsened performance is observed by both the algorithm and scheduler when there is a difference between the anticipated and actual variations in optimal stimulation parameters (Fig 9). This worsened algorithm performance can be compensated

using the smooth forgetting covariance function which maintains algorithm performance when there is a mismatch between the actual and anticipated temporal period by limiting the influence of the incorrect a priori information (i.e. the anticipated periodicity) on the TV-BayesOpt algorithm. Incorporation of both smooth forgetting and knowledge of the rhythmic variations in therapy efficacy leads to better performance of the TV-BayesOpt algorithm than when either of these are utilized by the algorithm alone.

To facilitate the development of next generation neuromodulation therapies, future work and consideration are required to implement the TV-BayesOpt algorithm on device hardware. These implementations may require the algorithm to be deployed on embedded devices that have limited computational and memory resources. In these cases, the performance of the algorithm must be further investigated to avoid performance degradation due to limited resources. Additionally, there are other questions related to the performance of the algorithm that should be further explored. Most notably, only relatively slow variations in the optimum stimulation parameter set were investigated in this present study to emulate temporal variations associated with biological rhythmicity and disease progression. The performance of the algorithm to track fast changes in the optimum parameter set which may be encountered during different activities of daily living or contexts was not evaluated. Moreover, the suitability of the algorithm for optimization over higher dimensional parameter spaces and whether further refinements to algorithm performance can be achieved by online hyperparameter optimization should investigated in future work.

## 5. Conclusion

In this study, a time-varying implementation of Bayesian Optimization was proposed to enable tracking of a dynamic, rhythmically varying optimal stimulation settings. The algorithm was tested in an oscillator model representative of oscillopathies and DBS. The TV-BayesOpt algorithm was utilized to track a temporally varying optimal stimulation phase for phase-locked DBS in the presence of both gradual and rhythmic (circadian or multidien rhythms) variations. The TV-BayesOpt algorithm demonstrated superior performance to standard, time-invariant BayesOpt resulting in lower regret over the optimization process. Finally, the forgetting factor, and the covariance hyperparameters were investigated which illustrated an interplay between these two hyperparameters leading to increased algorithm robustness. Future work is required to investigate implementations of the TV-BayesOpt algorithm in hardware.

## Supporting information

**S1 Fig. TV-BayesOpt algorithm performance for tracking a superimposed (gradual and periodic) drift in the optimal stimulation phase for phase-locked stimulation, $\psi^*$, in the presence of noise.** Panels A, C and E illustrate the tracking performance of the TV-BayesOpt algorithm (blue dots) at locating the true optimal phase value (black dots) for population desynchronization when no noise, Gaussian distributed noise with zero mean and a 0.75 standard deviation or Gaussian distributed noise with zero mean and a 1.5 standard deviation was added to the simulated optimum trajectory. Panels B, D and F illustrate the associated average regret for the TV-BayesOpt algorithm at tracking the true optimum phase value in comparison to when static BayesOpt was implemented alone for tracking the optimum trajectory in their associated Panels A, C and E.
(DOCX)

**S2 Fig. TV-BayesOpt algorithm performance for tracking a superimposed (gradual and periodic) drift in the optimal stimulation phase for phase-locked stimulation, $\psi^*$, for varying Kuramoto model parameter values.** Panels A, C, E and G illustrate the tracking performance of the TV-BayesOpt algorithm (blue dots) at locating the true optimal phase value (black dots) for population desynchronization when the Kuramoto model oscillator frequency was centered at 8 or 23 Hz. The frequency of oscillators in the underlying population were selected from a Gaussian distribution centered around each frequency. Panel C, E and G illustrate that increasing distribution of the oscillator natural frequencies, the size of the population or the natural frequency of the oscillator population does not affect algorithm performance at tracking the optimum stimulation phase since the shape of the objective is consistent across these parameters. Panels B, D, F and H illustrate the associated average regret for the TV-BayesOpt algorithm at tracking the true optimum phase value in comparison to when static BayesOpt was implemented alone for tracking the optimum trajectory in their associated panels A, C, E and G.
(DOCX)

## Author Contributions

**Conceptualization:** John E. Fleming, Oscar Lemmens, Angus Denison-Smith, Timothy Denison, Hayriye Cagnan.

**Formal analysis:** John E. Fleming.

**Investigation:** John E. Fleming, Ines Pont Sanchis, Oscar Lemmens, Angus Denison-Smith, Hayriye Cagnan.

**Methodology:** John E. Fleming, Ines Pont Sanchis, Timothy O. West.

**Software:** John E. Fleming.

**Supervision:** Timothy Denison, Hayriye Cagnan.

**Writing – original draft:** John E. Fleming, Ines Pont Sanchis, Hayriye Cagnan.

**Writing – review & editing:** John E. Fleming, Ines Pont Sanchis, Timothy O. West, Timothy Denison, Hayriye Cagnan.

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
