## [Decision Letter · Decision Letter 0]

17 Jul 2023

Dear Dr Cagnan,

Thank you very much for submitting your manuscript "From Dawn till Dusk: Time-Adaptive Bayesian Optimization for Neurostimulation" for consideration at PLOS Computational Biology.

As with all papers reviewed by the journal, your manuscript was reviewed by members of the editorial board and by several independent reviewers. In light of the reviews (below this email), we would like to invite the resubmission of a significantly-revised version that takes into account the reviewers' comments.

We cannot make any decision about publication until we have seen the revised manuscript and your response to the reviewers' comments. Your revised manuscript is also likely to be sent to reviewers for further evaluation.

Sincerely,

Daniele Marinazzo

Section Editor

PLOS Computational Biology

Daniele Marinazzo

Section Editor

PLOS Computational Biology

Reviewer's Responses to Questions

**Comments to the Authors: **

Reviewer #1: This work proposes a framework for more adequate parametrization of neuromodulation-based therapies. The main contribution is developing and evaluating a novel Bayesian optimization framework suitable for inferring dynamic deep brain stimulation timing (i.e. where the optimal phase is allowed to change in time). In principle, this allows for inference of optimal neuromodulation parameters as a function of time and time-dependent factors. The authors have motivated the need for such an extension with time-dependent factors such as disease progression and circadian rhythms but the approach is fairly general.

The approach is a time-varying extension of the conventional Bayesian optimization framework. The covariance function of the surrogate model in a standard setup is augmented and seen as a product of a spatial covariance (space between successive parameter estimates) and a temporal covariance term (capturing variability of the optimal parameters in time assuming no noise). The temporal covariance practically rescales the relevance of previously sampled data in relation to the most recent sample. This approach is sound and builds on related approaches to adaptive Bayesian optimization in Bogunovic et al. 2016 and Nyikosa et al. 2018 but with different assumptions about the temporal covariance. From methodological perspective, the developed approach is interesting since its temporal covariance combines terms which capture both periodic and gradual changes of the objective over time, going beyond simple stationary covariance matrix assumption. However, it does raise some minor questions I have regarding trade-off between the covariance hyperparameters mentioned below.

The proposed approach is studied in the context of parametrization of the Kuramoto model which is commonly workhorse in phase-locked deep brain simulation for treatment of oscillopathies. To recap, the Kuramoto model is parametrized with an intrinsic frequency parameter/parameters describing the oscillation frequency of the N coupled neurons, coupling term, intensity of the stimulation and a standard deviation of population oscillators from the natural frequency. The manuscript has summarized the fixed values of the Kuramoto model parameters adopted in all experiments where only the timing (i.e. phase of the stimulation) is inferred with Bayesian optimization. The choice of the Kuramoto model is a very well-motivated but additional background information on practical uncertainty associated with the hyperparameters (in particular the natural frequency and the stimulation intensity) would be beneficial. 

The authors consider a useful extension of the Kuramoto model to assume two distinct neuron categories based on their phase response curves: type I exclusive delay or advanced spike firing; type II both advance or delay dependent upon specific phase at which stimulation is delivered.

The objective function used in the proposed time-variant Bayesian optimization is the effect of the precise stimulation timing on minimizing the population synchrony. The performance evaluation then computes a cumulative regret using the known true values. This application specific evaluation criteria is arguably a lot more indicative of the practical utility of the proposed approach.

The optimal phase value for stimulation is considered to be the one leading to the greatest reduction in population synchrony. Some sensitivity analysis is offered to provide intuition in what different phase values are inferred when using alternative covariance structures (e.g. forgetting temporal covariance, periodic covariance), however, it was a bit unclear whether the forgetting term and the temporal period in the proposed forgetting periodic covariance are chosen independently. It seems to be the case that temporal period has been selected to capture circadian fluctuations (i.e. in 24 hour span) in the particular work, and then a range of values for the forgetting term are offered, but what are the practical trade-offs in achieving tractable inference was a bit unclear. 

Recommendation:

Overall the proposed framework for optimizing neuromodulation strategies dynamically offers an important contribution to the state-of-the-art.

Minor comments:

-It would be beneficial to comment on the computational consequences in terms of inference which following from replacing the stationarity of the covariance assumption in Nyikosa et al. 2022. 

-The authors did consider variation in the observed and assumed temporal drift, but it would be interesting to see how well the approach works when there are some deviations from the gradual drift assumed by the model in the generated data (e.g. adding different levels of noise during data generation) .

-Line 150 – ‘he’ is ‘The’

-The kernel equation in Figure 7A seems a bit out of place?

References:

Bogunovic I, Scarlett J, Cevher V. Time-varying Gaussian process bandit optimization.

Artificial Intelligence and Statistics. 2016;314–23

Nyikosa FM, Osborne MA, Roberts SJ. Bayesian Optimization for Dynamic Problems. arXiv Prepr [Internet]. 2018 Mar;arXiv:1803.03432. Available from:882 http://arxiv.org/abs/1803.03432

Reviewer #2: Authors consider a time-varying Bayesian optimization for tracking the level of synchronization in the Kuramoto-like model of phase oscillators. The final goal is to show that the algorithm can be used to track the optimum parameter set for neuromodulation therapy.

The results seem solid (I’m not an expert in statistics), but I’m still not convinced that they are generalizible enough to be used tor closed loop DBS, as authors claim. 

I would like to see if the procedure works for other type of oscillators, or even for spiking neurons, otherwise the work seems too theoretical. Similarly, it is not clear what is the reason for the choice of natural frequency, or why the number of oscillators is chosen to be so low. In the same sense, I’d imagine that the result of the model would be more realistic if there is also noise and even distribution of natural frequencies.

Minor comments:

- it is confusing that both spatial covariance and coupling are represented with K.

- Kuramoto order parameter is not defined. 

- why K and omega have subscripts biomarker, when biomarker is a measurable quantity, such as rho, while coupling and the natural frequency aren’t. 

- the representation of the periodic covariance function is also not typical. I

- maybe it is worth mentioning in the discussion that there is already some analytical results about Kuramoto model with time-varying parameters. Similarly the authors should specify that their algorithm is suppose to work only for slow-forcing of the coupling, when adiabatic reduction is also possible. The algorithm would probably fail for fast modulation of K.

**Have the authors made all data and (if applicable) computational code underlying the findings in their manuscript fully available?**

Reviewer #1: Yes

Reviewer #2: Yes

PLOS authors have the option to publish the peer review history of their article (what does this mean?). If published, this will include your full peer review and any attached files.

Reviewer #1: **Yes: **Y. P. Raykov

Reviewer #2: No
---

## [Decision Letter · Decision Letter 1]

9 Nov 2023

Dear Dr Cagnan,

We are pleased to inform you that your manuscript 'From Dawn till Dusk: Time-Adaptive Bayesian Optimization for Neurostimulation' has been provisionally accepted for publication in PLOS Computational Biology.

Best regards,

Daniele Marinazzo

Section Editor

PLOS Computational Biology

Daniele Marinazzo

Section Editor

PLOS Computational Biology

Reviewer's Responses to Questions

**Comments to the Authors:**

Reviewer #2: The authors have done a very nice job in thoroughly addressing all of my comments.

**Have the authors made all data and (if applicable) computational code underlying the findings in their manuscript fully available?**

Reviewer #2: None

PLOS authors have the option to publish the peer review history of their article (what does this mean?). If published, this will include your full peer review and any attached files.

Reviewer #2: No

---

## [Editor Report · Acceptance letter]

24 Nov 2023

PCOMPBIOL-D-23-00902R1 

From Dawn till Dusk: Time-Adaptive Bayesian Optimization for Neurostimulation

Dear Dr Cagnan,

I am pleased to inform you that your manuscript has been formally accepted for publication in PLOS Computational Biology. Your manuscript is now with our production department and you will be notified of the publication date in due course.

With kind regards,

Anita Estes
